

# Electric organ discharge from electric eel facilitates DNA transformation into teleost larvae in laboratory conditions

Shintaro Sakaki[1], Reo Ito[1], Hideki Abe[1], Masato Kinoshita[2], Eiichi Hondo[1] and Atsuo Iida[1]

[1] Nagoya University, Nagoya, Japan
[2] Kyoto University, Kyoto, Japan

## ABSTRACT

**Background:** Electric eels (*Electrophorus* sp.) are known for their ability to produce electric organ discharge (EOD) reaching voltages of up to 860 V. Given that gene transfer *via* intense electrical pulses is a well-established technique in genetic engineering, we hypothesized that electric eels could potentially function as a gene transfer mechanism in their aquatic environment.

**Methods:** To investigate this hypothesis, we immersed zebrafish larvae in water containing DNA encoding the green fluorescent protein (GFP) and exposed them to electric eel's EOD.

**Results and Discussion:** Some embryos exhibited a mosaic expression of green fluorescence, in contrast to the control group without electrical stimulation, which showed little distinct fluorescence. This suggests that electric eel EOD has the potential to function as an electroporator for the transfer of DNA into eukaryotic cells. While electric eel EOD is primarily associated with behaviors related to sensing, predation, and defense, it may incidentally serve as a possible mechanism for gene transfer in natural environment. This investigation represents the initial exploration of the uncharted impact of electric eel EOD, but it does not directly establish its significance within the natural environment. Further research is required to understand the ecological implications of this phenomenon.

## INTRODUCTION

The electric eel was originally described by Carl Linnaeus in 1766 as *Gymnotus electricus*, and later revised as *Electrophorus electricus* by Theodore Gill in 1864 (*Jordan, 1963*). In 2019, based on both morphological and molecular genetics, they were reclassified into three distinct species: *E. electricus*, *E. varii*, and *E. voltai* (*de Santana et al., 2019*). These creatures are renowned for their high-voltage electric organ discharges (EOD), which they employ for both predation and defense. In their natural environment, the maximum reported electrical current is 860 V in *E. voltai* (*de Santana et al., 2019*), with pulse durations of approximately 1 ms (*Catania, 2015*; *Crampton, 2019*). Additionally, three

Corresponding author
Atsuo Iida, tol2.4682@gmail.com

distinct types of EOD have been observed: (i) low-voltage pulses used for sensing, (ii) periodic high-voltage pulses generated while hunting in complex environments, and (iii) high-frequency volleys of high-voltage pulses during prey capture or as a defense mechanism (*Catania, 2014*). Electric eels predominantly hunt and capture prey, particularly fish, using the high-voltage EOD in their natural habitat (*Oliveira, Mendes-Júnior & Tavares-Dias, 2020*). In light of this background, we propose the hypothesis that the high-voltage EOD generated by electric eels during prey capture may have broader effects on neighboring organisms and their habitats in their natural environment.

Electroporation is a well-established gene transfer technique in molecular biology, involving the application of an electric field to introduce exogenous DNA into bacteria, cultured cells, or multicellular organisms (*Shi et al., 2018*; *Calvin & Hanawalt, 1988*; *Chu, Hayakawa & Berg, 1987*; *Nakamura et al., 2004*). However, the use of electroporation is typically confined to laboratory settings and is not observed in natural conditions. While the presence of electricity in natural organisms is widely documented, its ecological roles remain largely enigmatic (*England & Robert, 2022*). For instance, high-voltage discharges can naturally occur, such as in lightning strikes from thunderclouds. A previous study has proposed the possibility of lightning strike-driven DNA transformation and horizontal gene transfer among environmental organisms (*Kotnik, 2013*), but this hypothesis has not yet been experimentally verified.

In this study, we examined the potential for electric eel-induced gene transfer by exposing zebrafish larvae to the electric eel's EOD, while they were incubated with plasmid DNA encoding GFP. While artificial electroporation methods for zebrafish have been extensively established in laboratory conditions, we aim to investigate the feasibility of natural electroporation facilitated by electric eels. The optimal electrical parameters for machine-driven electroporation in zebrafish have been previously documented for various developmental stages, including fertilized eggs, larvae, and adult fish (*Cerda et al., 2006*; *Iwaizumi, Yokoi & Suzuki, 2020*). In the study, we reared electric eels in our laboratory and exposed zebrafish larvae to their EOD while they were incubated with the DNA solution.

## MATERIALS AND METHODS

### Animal experiments

This study was approved by the Ethics Review Board for Animal Experiments at Nagoya University (approval number A210748-002). We conducted the experiments using live animals, ensuring that they were under anesthesia with 0.01% tricaine, in strict accordance with the institutional guidelines. In cases where the animals experienced severe health deterioration during the experiment, we euthanized them under ice anesthesia with 0.01% tricaine.

### Fish maintenance

The electric eel (*Electrophorus* sp.) was purchased from Meito Suien Co., Ltd (Nagakute, Japan). The fish was housed in a freshwater tank (90 × 45 × 45 cm) at 27 °C under a 14 h:10 h light: dark photoperiod cycle. To regulate the pH of the tank water within the range of 7.0–8.0, approximately 1.5 g of sodium hydrogen carbonate was added to the fish

tank once a week. They were fed a pellet diet (Meito Suien, Namazu-Gohan M) or live goldfish or Japanese loach.

The Riken-Mic zebrafish strain was utilized as the recipient for electroporation. Adult zebrafish were housed in a circulation water tank designed for small fishes (Meito system) and fed a powder diet or live Artemia. Fertilized eggs and embryos were maintained in 0.3% sea salt water.

## Plasmids

The donor DNA used for electroporation was an 8,888-bp plasmid containing the GFP coding sequence driven by the *Oryzias latipes β-actin* promoter (accession number LC775392). Plasmids were amplified in DH-5α competent cells (TaKaRa, #9057) and subsequently extracted and purified using the Plasmid DNA Extraction Midi Kit (FAVORGEN, #FAPDE 002). The DNA solutions were stored at −20 °C until use.

## Devices to record and analyze electric pulses

The electric eel's discharge waveform was recorded using the following method. Recording and reference electrodes were constructed by connecting a 99.9% carbon rod (diameter: 10 mm, length: 100 mm) to a lead wire. One of these electrodes was positioned in an experimental tank (60 × 30 × 36 cm) containing a single electric eel, and it served as the reference electrode while being grounded. Additionally, one of the two electrodes was placed on the electric eel's head as the positive pole, and the other carbon electrode was positioned on its tail as the negative pole. The voltage between the two electrodes was attenuated to 1/500 of the original signal using a High Voltage Differential Probe (Micsig Technology, #DP10013) and then input into a USB Data Acquisition Module (Data Translation, #DT9812-10V), which was connected to a Windows PC for display and recording. The sampling frequency was set to 20 kHz. The waveform recording and subsequent analysis were conducted using the DataView software (https://www.st-andrews.ac.uk/~wjh/dataview/).

## Electroporation

The electric eel was relocated to the experimental tank, which was equipped with electrodes, a grounding system, and a computer containing the USB Data Acquisition Module (as depicted in Fig. 1A). The tank was filled with approximately 50 liters of freshwater maintained at 27 °C. The 6-day post-fertilization zebrafish larvae were incubated overnight in a 1 ng/μl plasmid solution diluted in 0.3% sea salt water. Subsequently, the larval samples were transferred to Electroporation Cuvette (#5510-11: Thermo Fisher Scientific, Waltham, MA, USA) containing a fresh 1 ng/μl plasmid DNA solution. Each cuvette accommodated a maximum of ten larvae, and the requisite number of cuvettes for each trial was attached to the electrode. This approach enabled exposure of more than ten larvae to an equivalent electric discharge in a single trial. The cuvettes were submerged in the experimental tank, and the electric eel initiated EOD during prey capture by feeding on an anesthetized goldfish. The goldfish was secured in the electrode using clips and a 6 cm thread, positioning the bait fish within 6 cm of the electric eel's head, as
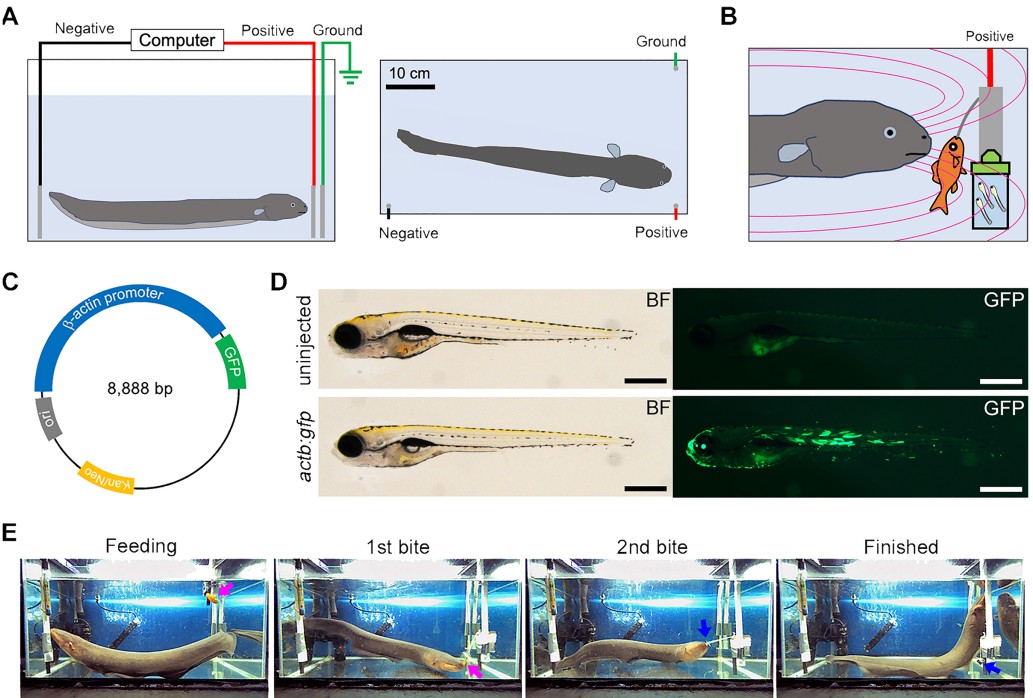

**Figure 1 EOD exposure from electric eel to zebrafish larvae.** (A) This illustration depicts the experimental tank used to expose the recipient organism to the electric eel's electric organ discharge (EOD). Within the tank, three carbon rod electrodes are placed: two inputs (colored black and red) and a ground electrode (colored green). (B) An EOD is induced by the electric eel during its predatory behavior when it feeds on a goldfish. The cuvette containing zebrafish larvae and a DNA solution is positioned in close proximity to the high-voltage pulses generated by the electric eel. The magenta curve illustrates the electric field produced by the electric eel. (C) This illustration indicates the construction of the GFP expression plasmid driven by the *Oryzias latipes actb* promoter. (D) The photograph displays 7-day post-fertilization (dpf) *Danio rerio* (zebrafish) larvae that were subjected to microinjection with the indicator plasmid at the one-to-eight cell stages. The plasmid has resulted in robust and widespread GFP fluorescence without apparent developmental abnormalities. Scale bar, 500 μm. (E) An example of a single predatory behavior with the EOD pulses lasting 30 s is shown. The electric eel first bites and swallows the goldfish (indicated by the magenta arrow, 1st bite), followed by another bite at an empty clip (indicated by the blue arrow, 2nd bite). The full video is available as Movie S1.

illustrated in Fig. 1B. The experiment duration, from submerging the cuvettes to the final retrieval, was maintained at approximately 30 s for each trial. In most of the trials, the electrical discharge during prey capture was completed within this 30 s timeframe. Control groups were established without EOD, DNA, or both components. Even under the no electrical stimulation condition, the larvae were placed in the cuvettes and kept for the same duration as the EOD manipulation.

## Fluorescent imaging

For fluorescent microscopy, zebrafish larvae that were 1-day post-electroporation were placed on 1.0% agar substrate within Petri dishes. Images were acquired using a Leica M165FC microscope (Leica Microsystems) equipped with a suitable filter set (GFP, #10447408; RFP, #10447417). Both brightfield and fluorescent images were captured using

a Leica K3C digital color camera, and the imaging process was facilitated with Leica Application Suite (LAS) software. Following the microscopy, the zebrafish larvae that had survived were euthanized using ice anesthesia with 0.01% tricaine solution.

## Statistical calculation

The data is presented as the mean ± standard error of the mean. Statistical analyses were conducted using the freely available and open-source software, Jeffreys's Amazing Statistics Program (JASP), which can be accessed at https://jasp-stats.org/. A *p*-value below 0.05 was regarded as statistically significance.

# RESULTS

## Electric eel EOD exposure to zebrafish larvae

In this study, we employed an electric eel (*Electrophorus* sp.; gender unknown; approximately 60 cm in total length) as the electric source for our investigation (Fig. 1A). To set up the system, we arranged a test tank with electrodes and a computer. The electric eel was fed a goldfish to elicit the EOD, and high-voltage pulses were exposed to the zebrafish embryo in an electroporation cuvette field with a DNA solution at close range (Fig. 1B). We used a GFP expression plasmid driven by the *O. latipes β-actin* (*actb*) promoter as a transgenic indicator (Fig. 1C). Prior to the experiment, we confirmed that the *actb:gfp* cassette induced strong GFP fluorescence in zebrafish larvae (Fig. 1D). The electric eel emitted EOD waveforms when it received an anesthetized goldfish as prey (Fig. 1E, Movie S1). This waveform was characterized by a sequence of distinct high-voltage pulses synchronized with the eel's bite on the prey (Fig. 2A). During the bite, the EOD waveform comprised a series of several to approximately 10 high-frequency pulses (200–400 Hz), followed by low-frequency pulses (about 50 Hz) (Fig. 2B). This waveform pattern was repeated several to about 10 times with individual bites.

The recorded individual EOD pulse waveform exhibited a monopolar (DC), head-positive bell shape (Fig. 2C) during a single bite, which is different from the square pulses produced by machine electroporators typically used in electroporation into organisms (Figs. 2D and 2E). The amplitude of the pulses measured 184.6 ± 1.9 V (Fig. 2F), with a half-width of 0.501 ± 0.001 ms (*n* = 447; Fig. 2G). The intervals between the EOD pulses varied, with a median of 13.8 ms (mean ± se = 19.4 ± 2.1 ms; Fig. 2H). Within approximately 10 s, the electric eel generated around 500 pulses during a single bite. This pattern was repeated approximately 2-3 times during a single feeding behavior. As a result, in electroporation using EOD, on average, bell-shaped pulses were applied 1,000–1,500 times within 30 s. Although the number of pulses per bite, amplitude, and intervals differed in each trial or eel's behavior, the half-width remained consistent and reproducible.

## GFP fluorescence in zebrafish larvae after EOD exposure

At the outset, we attempted to employ fertilized eggs at the cleavage stages or 1- to 2-day post-fertilization (dpf) embryos for electroporation driven by electric eel discharges. Regrettably, these embryos did not survive, primarily due to yolk sac rupture during electric exposure or other manipulations. Consequently, we opted to utilize 7-dpf embryos,

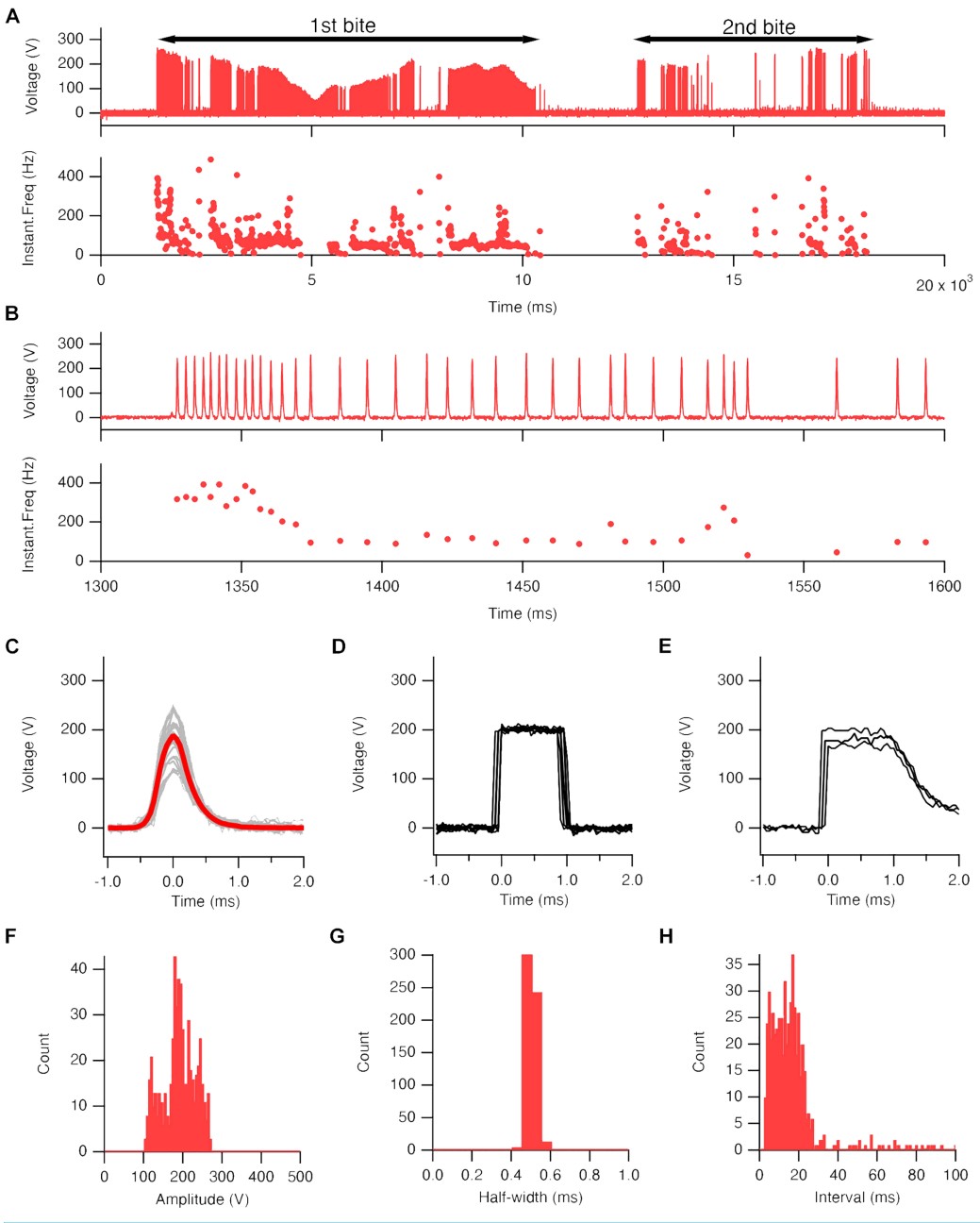

**Figure 2 Electric eel EOD waveform.** (A) An example of the recorded EOD waveform (upper trace) and its instantaneous frequency changes (lower trace) during a single predation behavior (Movie S1). Black bars on the trace represent the duration of the electric eel bite of the goldfish. (B) An enlarged view of trace A. (C–E). Superimposed 29 traces (gray line) and their average (red line) of high-voltage EOD pulse (C), output waveform of CUY21EDITII (D), and NEPA21 machine electroporators (E). (F and H) Amplitude (F), half-width (G), and pulse interval (H) of EOD pulses (447 pulses from single fish).

which had advanced beyond the yolk sac stages and exhibited a more robust body structure. In each trial, we subjected 20–49 larvae to the plasmid DNA solution using distinct electric eel discharges, conducting a total of fifteen repetitions (Table S1).

2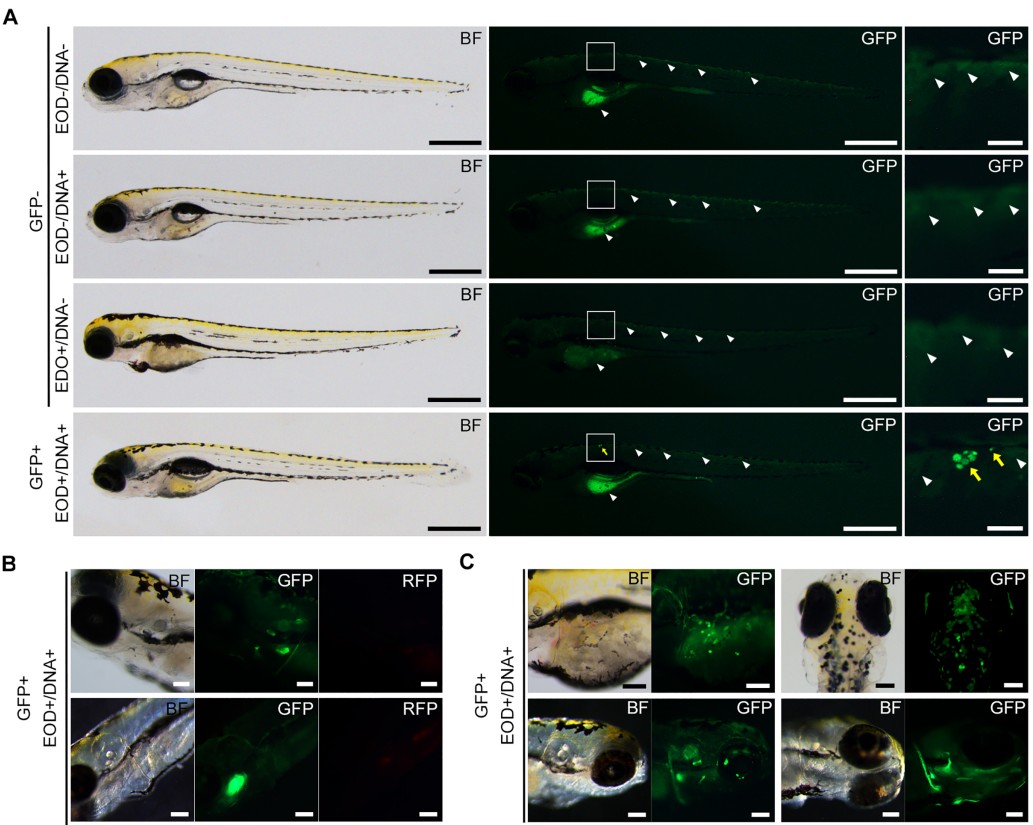

**Figure 3 GFP fluorescence in EOD-exposed zebrafish.** (A) These photographs showcase 8-day post-fertilization larvae. In the EOD-/DNA-, EOD-/DNA+, and EOD+/DNA- panels, representative images of transgenic GFP-negative specimens are depicted. The white arrowheads point to background autofluorescence observed in the yolk and dorsal skin of samples manipulated without DNA. The EOD+/DNA+ panel displays representative images of transgenic GFP-positive specimens, taken 1 day after EOD exposure. The yellow arrows highlight GFP fluorescent cells, distinguishable from autofluorescence. Scale bars, 500 µm (entire body) and 100 µm (enlarged views). (B) Fluorescent observation of the region of interest (ROI) containing GFP-positive cells was performed using an RFP filter. The GFP-positive signals are clearly differentiated from the RFP signals. Scale bar, 100 µm. (C) This image shows additional specimens of GFP-positive larvae following EOD exposure with DNA. Clusters of fluorescent cells are distributed on the dorsal skin, yolk surface, and cephalic region, including the eye. Scale bar, 100 µm.

Following these experiments, all specimens were subjected to examination under a stereomicroscope, and their survival rates and the extent of green fluorescence expression were assessed by a blinded evaluator. In our experimental conditions, no apparent damage was observed in the surviving specimens after the EOD exposure. The body shapes of these specimens closely resembled those of non-EOD-exposed specimens. Upon visual inspection, weak fluorescence was noted in the ventral yolk and dorsal skin of most zebrafish larvae, with more pronounced fluorescence detected in some of the samples (Fig. 3A). Occasionally, even in the absence of DNA, isolated single cells exhibited green fluorescence. Such signals were categorized as autofluorescence or false positives and were excluded from being considered as positive in all conditions. Even when multiple instances of green fluorescence were identified, signals overlapping with red fluorescence were not

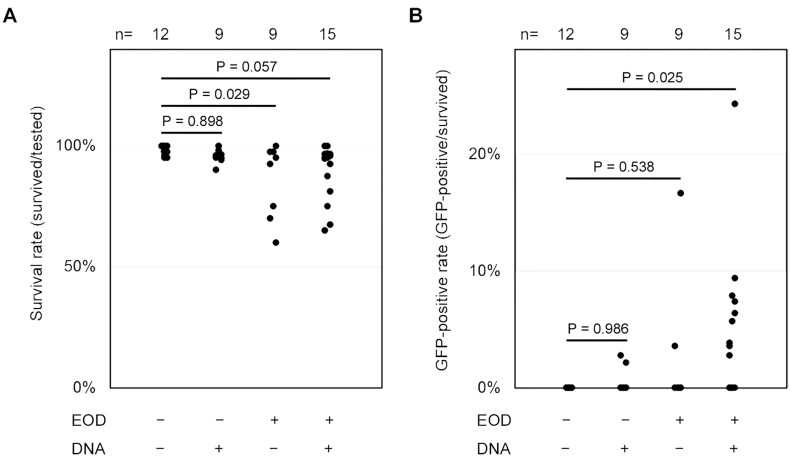

**Figure 4 Statistical analyzes of survival and GFP-positive rate.** (A and B) These scatter plots depict the distributions of the survival rate (A) and GFP-positive rate (B) per trial. The p-values were determined using Dunnett's test. The raw data for these distributions can be found in Table S1. n, number of trials.

included in the count (Fig. 3B). To distinguish these background signals or false positives from the expected GFP signal, our focus was on clusters of multiple cells displaying intense green fluorescence while lacking red fluorescence. These clusters of GFP-positive cells were primarily observed on the body's surface, including the trunk skin, yolk surface, eyes, and other cephalic tissues (Fig. 3C). Based on this criterion, the survival rate and GFP-positive rate were compared among the four conditions, EOD-/DNA-, EOD-/DNA+, EOD +/DNA- and EOD+/DNA+. Despite some trials in the EOD+/DNA- condition appearing to exhibit a decreased survival rate, the differences observed were not statistically significant according to the Dunnett's test (mean ± sd = 98.4 ± 2.1% for EOD-/DNA- ($n$ = 12) $vs.$ 96.0 ± 3.1% for EOD-/DNA+ ($n$ = 9), $P$ = 0.898; EOD-/DNA- $vs.$ 87.2 ± 14.8% for EOD+/DNA- ($n$ = 9), $P$ = 0.029; EOD-/DNA- $vs.$ 89.6 ± 11.4% for EOD+/DNA+ ($n$ = 15), $P$ = 0.057; Fig. 4A). Importantly, 88% of the samples survived until 1 day after the manipulation, even in the EOD+/DNA+ condition (468/532; Table S1). The frequency of larvae exhibiting GFP fluorescence in the EOD+/DNA+ condition was significantly different from the EOD-/DNA- condition according to the Dunnett's test (mean ± sd = 0.0 ± 0.0% for EOD-/DNA- ($n$ = 12) $vs.$ 0.5 ± 1.1% for EOD-/DNA+ ($n$ = 9), $P$ = 0.986; EOD-/ DNA- $vs.$ 2.3 ± 5.5% for EOD+/DNA- ($n$ = 9), $P$ = 0.538; EOD-/DNA- $vs.$ 4.8 ± 6.4% for EOD+/DNA+ ($n$ = 15), $P$ = 0.025; Fig. 4B). In total, 5.3% of the survived EOD+/DNA+ larvae exhibited GFP-positive cells (25/468; Table S1).

## DISCUSSION

In this study, we propose that electric eel EOD facilitates the transportation of DNA into cells from the surrounding water environment. This suggests that electric-based transgenesis might occur not only through artificial electroporators in laboratory settings, but also through living electric organisms in their natural habitats. Previous reports have indicated the presence of 2.68 ng/μl of environmental DNA (eDNA) extracted from the Amazon hydrosphere (*Bevilaqua et al., 2020*). The eDNA from the Amazon River has been

used for identification of fish species in the hydrosphere (*de Santana et al., 2021*; *Batista et al., 2022*). These findings support the notion that the hydrosphere of the Amazon River contains the necessary components for natural electroporation, including the electric eel as a power source, live fish as recipients, and eDNA as a source of genetic factors. We hypothesize that horizontal gene transfer may occur among organisms, including fish, in the habitat surrounding electric eels.

Our demonstration solely pertains to environmental gene transduction in somatic cells, and we have not yet confirmed whether the transgene functions as a heritable factor in offspring. In the case of electroporation using machinery, some studies have reported the introduction of exogenous genes into the germline cells of vertebrates, including fish (*Ohtsuka et al., 2018*; *Widłak et al., 2003*; *Sin et al., 2000*). Under various natural conditions, it is not impossible for environmental factors to align with optimized conditions for gene transfer into the germ cells of organisms. We hypothesize that heritable transgenesis mediated by electric organisms could occur by chance if given sufficient time in natural conditions.

To validate heritable transgenesis, unicellular organisms are suitable models because they reproduce the next generation through division or budding. We investigated electric eel-based gene transfer using the unicellular organisms *Escherichia coli* and *Paramecium* sp (*Dower, Miller & Ragsdale, 1988*; *Kung et al., 2000*). However, our experiments did not yield positive results, suggesting that electric eel EOD may not facilitate the transfer of genetic markers into these cells effectively. For *E. coli*, machine-based electroporation typically employs discharges exceeding 1 kV cm$^{-1}$ (*Calvin & Hanawalt, 1988*; *Dower, Miller & Ragsdale, 1988*). In our experimental setup, the voltage generated by the electric eel was approximately 200–250 V, which might have been insufficient for effective electroporation into *E. coli* cells. Previous reports have indicated that *Paramecium* can uptake exogenous DNA through electroporation (*Boileau et al., 1999*). Therefore, we attempted to electroporate *Paramecium* using established plasmid DNA carrying GFP (*Takenaka et al., 2002*). Regrettably, we could not replicate DNA electroporation into *Paramecium* under our experimental conditions. In a previous study, electroporation into *Paramecium* was achieved successfully using a discharge of 500 V cm$^{-1}$ (*Boileau et al., 1999*). Hence, it is plausible that the voltage generated by the electric eel in our study may be relatively lower or inadequate for the same purpose. Continued validations using unicellular organisms are necessary to further support our hypothesis that electric eels facilitate heritable transgenesis in animals.

It is important to consider the differences in pulse shapes between electric eel-based electroporation and machine-based electroporation. In many instances, machine electroporation conditions are optimized using square pulses to achieve maximum viability and efficiency (*Heiser, 2000*; *Jordan et al., 2008*). However, the pulses recorded from our electric eel exhibited bell-shaped waveforms, which differs from square pulses commonly used in machine-based electroporation. While the efficiency of natural electroporation cannot be solely predicted based on the pulse shape, it's possible that the variations between previous reports utilizing machine electroporators and our study employing electric eels can, in part, be attributed to these differences in pulse shape.

We carefully designed the experiment to ensure the consistency of the discharge conditions received by the zebrafish larvae in each trial. However, it's worth noting that factors such as the pulse shape during prey capture and the precise distance and location between the electric eel and the electroporation cuvette may not be entirely uniform across all trials. To address this issue, we acknowledge the need for improved experimental settings that can minimize variations between trials. One potential approach could involve the use of a machine electroporator capable of delivering consistent and repeatable pulses, similar to the electric eel's EOD. Additionally, we recognize the potential suitability of employing a generalized linear mixed model as a statistical method in future investigations to account for variability between trials (*Breslow & Clayton, 1993*; *Bolker et al., 2009*).

In our study, we observed a few cases of false positive fluorescence in samples that were exclusively exposed to the EOD or DNA. These signals may encompass autofluorescence in cells damaged by the electric discharge or other factors (*Surre et al., 2018*; *Kozlova et al., 2020*). In samples treated only with the plasmid DNA, we cannot exclude the possibility that the observed fluorescence indicates spontaneous DNA incorporation into the cells (*Tsoncheva et al., 2005*). While we present a novel finding that electric eel EOD enhances genetic transformation in fish larvae, the underlying mechanisms remain unclear and necessitate further investigation.

## CONCLUSIONS

This study offers evidence supporting the idea that electric discharge from electric eels can enhance DNA transformation in zebrafish larvae. Our innovative approach, utilizing live animals as the source of electric discharges, suggests the potential for such discharges to contribute to gene transfer in natural environments. Nevertheless, it is essential to recognize that our research was conducted in controlled laboratory conditions. Therefore, we cannot definitively assert that electric discharges act as a genetic factor in natural settings based solely on our findings. Further investigations are necessary to explore the heredity of electric discharge-mediated transgenesis and to accumulate evidence for its occurrence in natural habitats.

## ACKNOWLEDGEMENTS

We thank Yukimasa Onabe for his support in arranging the electric eel. Ryo Sasaki helped to set up the microscopy system. Yasuhiro Takenaka kindly shared the DNA samples for Paramecium. We are grateful for the helpful discussions with Akihiko Koga and members of the Unique-Kai, an annual meeting for biologists investigating non-conventional experimental animals in Japan.

### Funding

This work was performed with a laboratory budget from Nagoya University. The funders had no role in study design, data collection and analysis, decision to publish, or preparation of the manuscript.

## Grant Disclosures

The following grant information was disclosed by the authors:
Nagoya University.

## Competing Interests

The authors declare that they have no competing interests.

## Author Contributions

- Shintaro Sakaki conceived and designed the experiments, performed the experiments, analyzed the data, prepared figures and/or tables, and approved the final draft.
- Reo Ito performed the experiments, analyzed the data, authored or reviewed drafts of the article, and approved the final draft.
- Hideki Abe performed the experiments, analyzed the data, prepared figures and/or tables, authored or reviewed drafts of the article, and approved the final draft.
- Masato Kinoshita performed the experiments, authored or reviewed drafts of the article, and approved the final draft.
- Eiichi Hondo conceived and designed the experiments, authored or reviewed drafts of the article, and approved the final draft.
- Atsuo Iida conceived and designed the experiments, performed the experiments, analyzed the data, prepared figures and/or tables, authored or reviewed drafts of the article, and approved the final draft.

## Animal Ethics

The following information was supplied relating to ethical approvals (*i.e.*, approving body and any reference numbers):

The Ethics Review Board for Animal Experiments at Nagoya University approved the study (A210748-002).

## Data Availability

The raw data are available in the Supplemental Files.

## Supplemental Information

Supplemental information for this article can be found online at http://dx.doi.org/10.7717/peerj.16596#supplemental-information.

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
