# Peer review of "Electric organ discharge from electric eel facilitates DNA transformation into teleost larvae in laboratory conditions"

_PeerJ, doi:10.7717/peerj.16596_

## Round 0.1 · original submission · Major Revisions

Thank you for submitting your manuscript to PeerJ. The two reviewers of your paper were positive about its contributions but identified several concerns and made a number of suggestions that I invite you to address.

Reviewer 1 posed questions about details in the methods. They also suggest including additional images of representative fish for two additional conditions in Figure 3. The reviewer has a suggestion for how to better distinguish between autofluorescence and GFP expression using a longer wavelength filter. I recognize that this would require the re-collection of data from very large samples size. Perhaps you can further describe the distinction between autofluorescence and GFP expression in the manuscript, or explain in your rebuttal letter how you are confident in the distinction. A similar point is raised by reviewer 2.

Reviewer 2 was very positive about the novelty and design of your study and data presentation, however they raise a number of issues for you to consider. They ask several questions about the experimental design, including whether zebrafish were scored blind and asked for references to support your distinction between single-cell and autofluorescence and a true GFP signal. The reviewer also makes some valuable suggestions regarding interpretation of the results so that the reader is aware of possible limitations of how your findings may relate to possible mechanisms in the wild. While these caveats are made clear in your discussion, the reviewer suggests that they be clearer in the title and abstract.

I invite you to submit a revised manuscript after major revisions and ask that your rebuttal letter address all reviewer comments.

I look forward to receiving your revised submission.

·

Basic reporting

Clear, unambiguous, professional English language used throughout. - True
Intro & background to show context. Literature well referenced & relevant. - True
Structure conforms to PeerJ standards, discipline norm, or improved for clarity. - True
Figures are relevant, high quality, well labelled & described. - True
Raw data supplied (see PeerJ policy). - True

Experimental design

Original primary research within Scope of the journal. - True
Research question well defined, relevant & meaningful. It is stated how the research fills an identified knowledge gap. - True
Rigorous investigation performed to a high technical & ethical standard. - True
Methods described with sufficient detail & information to replicate. – Mostly true

On lines x-y the authors describe using a rice fish promoter to drive GFP. There should be an description of the sequence of the plasmid or other identifier so that other researchers can replicate this reagent. There should also be a citation to the research validating the plasmid.

I am unable to determine what software, if any, was used to calculate p values of X^2 analyses.

Validity of the findings

Although unnecessary for PeerJ, the findings are novel. Rationale and benefit to science are clear.
All underlying data have been provided; they are robust, statistically sound, & controlled. - True
Conclusions are well stated, linked to original research question & limited to supporting results. - True

Additional comments

Complements
1) The descriptions of the waveform of the eel are much appreciated.

2) Descriptions of the electroporation rig are well done.

Issues that could be corrected
1) It would be best to show representative pictures of fish that were counted positively for fluorescence under all conditions. Missing only are EOD-/DNA+ and EOD+/DNA- conditions. The most important fish are shown, namely EOD-/DNA- and EOD+/DNA+.

2) Observation of GFP expressing cells using a filter allowing longer wavelengths, rather than the filter used in this study, may distinguish GFP expressing cells from autofluorescing cells and eliminate the ambiguity of false positives found in the study in the EOD-/DNA+ and EOD+/DNA- conditions.

3) A minor typo on line 111 should read: electroporation, ten larvae…

Reviewer 2 ·

Basic reporting

The quality of English language in this manuscript is generally good, however there are some grammatical errors that need to be addressed. For example, “sp.” should not be italicised. Care should also be taken to make sure that “electric organ discharge” and “electric eel” are either preceded by “the” or used in their plural forms as appropriate. There are other grammatical mistakes throughout, but these are the most frequent ones. This should be rectified before publication but does not significantly impact comprehension of the article. I would say the literature cited is sufficient to build the background, although perhaps some more references to the wider ecological influence of electricity, for example England & Robert 2022 (Biological Reviews) and some of the references therein would be beneficial. The article is well-structured and clear, and the figures are beautiful and easy to interpret. The raw data have been shared. The hypothesis of the study is very unique, and the study is self-contained with results relevant to the hypothesis.

Experimental design

This research is very original and presents primary data. It lies within the scope of PeerJ. The research question is clearly defined and interesting, although exactly how meaningful it is remains to be seen, as I outline in more detail in Section 3 of this review. In terms of the methodology, this is generally well-reported but some more detail is needed in places. The statistics need to be reported more formally, including chi-squared values. Perhaps an alternative statistical test or interrogation could be used though, so that the “- -“ group can also be included in the analysis. I also would like to know if the experimenter judging the fluorescence was blind to the treatment conditions. If not, I would recommend re-running the analysis with a blind scorer. I would also appreciate some more references to the literature showing that the criteria by which single-cell fluorescences were excluded, and that a fluorescence rate of 5% is meaningful and widely accepted as a strong signal or not. Can the authors also provide some more detail, even estimations, of the distances between the eel and the zebrafish? The sample sizes are nice and high, and appear robust – but multiple zebrafish were exposed to the same EOD at the same time. How was this factor addressed statistically? I think more detail is needed in the text to describe both the methods and results of each treatment/control condition, as this is not immediately clear from the text alone. It is much clearer in the figures though, which look great.

Validity of the findings

It is a very cool finding, however, as the authors point out in the conclusions, this data cannot be fairly used to support the idea that electric eels facilitate electroporation in nature. Based on this data, conclusions can only be realistically made for this process under laboratory conditions. This is a very important distinction to make, and I applaud the authors for making it clearly in that section, but I think it needs to be made earlier in the manuscript too. I’m very concerned that based upon the title and abstract of this manuscript, that the nuance of the finding will be lost, and the media and wider scientific community will incorrectly assume that this phenomenon has been proven to take place in nature. I would therefore highly recommend the title and earlier sections of the manuscript are modified to make this distinction clearer. The meaningfulness of the finding is further brought into question by the fact that fatality and severe injury were regularly seen in the zebra fish subjects, which brings into question the extent to which this phenomenon could be evolutionarily incorporated. Furthermore, the results could not be replicated in unicellular organisms. Could the authors comment on reasons why it may have worked in zebrafish but not these other organisms? Beyond these points, I think some qualification of the extent to which this could be an evolutionary driver, even if it does occur in nature, is needed. Electric eels are relatively rare and not geographical widespread, therefore the impact this process could have on wider evolutionary processes is incredibly limited. I don’t think this makes the findings uninteresting, but perhaps this caveat should be stated in the manuscript. Overall though, this is a very interesting and well-presented study, and the conclusions are stated clearly and linked.

---

## Round 0.2 · Minor Revisions

Thank you for submitting your revised manuscript to PeerJ and for your response to reviewer comments. Reviewer 1 is happy with your revisions and only points out one grammatical issue in line 187 that they suggest correcting in the final version. Reviewer 2 writes that many of their concerns were addressed but asked that you consider two points. First are some suggestions for editing for grammatical clarity, and suggested that you review your newly added language as well. Second, they have remaining questions about the statistical analysis that I would like to ask you to address before the manuscript can be accepted.

Please consider the remaining concerns of reviewer 2 in a revised manuscript and rebuttal letter.

I look forward to receiving your revised submission.

·

Basic reporting

Clear, unambiguous, professional English language used throughout. - True
Intro & background to show context. Literature well referenced & relevant. - True
Structure conforms to PeerJ standards, discipline norm, or improved for clarity. - True
Figures are relevant, high quality, well labelled & described. - True
Raw data supplied (see PeerJ policy). - True

Experimental design

Original primary research within Scope of the journal. - True
Research question well defined, relevant & meaningful. It is stated how the research fills an identified knowledge gap. - True
Rigorous investigation performed to a high technical & ethical standard. - True
Methods described with sufficient detail & information to replicate. – True

Validity of the findings

Although unnecessary for PeerJ, the findings are novel. Rationale and benefit to science are clear.
All underlying data have been provided; they are robust, statistically sound, & controlled. - True
Conclusions are well stated, linked to original research question & limited to supporting results. - True

Additional comments

There remains a typo that I hope the authors can clear up before publication.
Lines 186-7 currently read “ 186… While some trials of the EOD+/ DNA- condition exhibited a decrease in the 

187 survival rate, it was no statistical differences by…” 

Might I suggest it read as follows.
… While some trials of the EOD+/DNA- condition appeared to have decreased survival rate, there was no statistical difference by the Dunnett’s …

Reviewer 2 ·

Basic reporting

My questions and concerns have been addressed here, although for grammatical clarity I would recommend using "conditions" in the title rather than "condition". Also the added sentence "The electricity of nature organisms has been various reported, but the roles on the ecological scale remain enigmatic", could better worded as "The electricity of natural organisms has been widely reported, but its roles on the ecological scale remain enigmatic". Some other tidying up of the English is needed, especially in the newly added sentences.

Experimental design

My concerns and questions here have been answered.

Validity of the findings

Most of my questions and concerns have been addressed here, although I still have some unresolved queries about the statistical techniques. I raised the point that multiple (10 per trial in the methods but 20-40 per trial in the results - is this a contradiction?) fish were exposed to the same EOD at the same time, meaning that these fish cannot be considered statistically independent from one another, but appear to have been treated as such from the sample sizes given. I don't believe the use of Dunnett's test resolves this issue. If this has not been justified or accounted for statistically it may be a case of inadvertent pseudoreplication? But either way I do not feel as if this aspect of the study is clear.

---

## Round 0.3 · Minor Revisions

Thank you for all of your work on the revisions. Reviewer 2 notes that the article is well presented and interesting. The reviewer still expresses some concerns about clarity in the methods regarding whether multiple cuvettes are exposed simultaneously to the same pulses from a single eel feeding event. However, I believe that your description in the Methods lines 117-120 makes this clear. The reviewer also notes that the methods do not mention whether the non-EOD treatments were kept in the cuvettes for the same length of time as those exposed to the electrical pulses. That may be a detail worth adding to the methods section. Also, did treatment length vary between the EOD trials?

The reviewer also expressed concerns about the potential for pseudoreplication and the way the statistical method accounts for differences in pulse characteristics from different EOD trials. I believe that the added detail in your methods makes clear that each individual larva in a treatment is not being treated as an individual sample, which avoids the pseudoreplication concern. But the reviewer points out that variation in pulse characteristics between treatments is not accounted for in the statistic used. They suggest a statistical approach that would look for the effects of trial and cuvette number. If you decide to stay with the Dunnett’s test currently used in the manuscript, I suggest adding a comment in the discussion section to point out the possible complication of variation between pulses in the EOD trials so readers are aware of this additional variable.

I noticed a few areas that could use some additional editing:

Line 30: Change to “showed little distinct fluorescence”

If you use a structured abstract I would change the last section to “Results and Discussion”

Line 50: Change to “. . . EOD during prey capture causes further effects on the neighboring . . .”

Line 76: “When the animals severely undermine their health . . . “ is awkward as written.

Line 219 – Change to “were compared”

Line 244/245: Change to “However, the pulses recorded from our electric eel were bell shaped.”

I look forward to receiving your revised submission.

Reviewer 2 ·

Basic reporting

The authors have improved the English even further and the article reads well.

Experimental design

The authors have made the methods clearer but I'm still uncertain as to whether larvae with different DNA conditions were exposed to the same EODs? Or was every cuvette in a specific trial all one condition? Furthermore, in the conditions where no EOD was applied, were the larvae kept in the cuvettes/DNA solution for the same amount of time as those exposed to EODs?

Validity of the findings

I appreciate the added detail in the methods but I do not believe these details alleviate my concerns around the statistical validity of the findings. My issue is that the EODs were not identical between each trial, therefore these differences in EOD strength/frequency/waveform need to be taken into account. If multiple larvae were exposed to the same EOD, and multiple larvae were exposed to a different EOD, then the larvae within each trial cannot be considered statistically independent. My suggestion is that the authors redo the statistics in a way that controls for trial number and perhaps cuvette identity. For example, you could used Generalized Linear Mixed-Effects Models, with a binary data distribution specifier, and have trial number and cuvette identity as random effects in each of your models. I think this would solve the problem, but as it stands, I do not think the statistics currently described are appropriate for the methods as I understand them.

Additional comments

The article is otherwise really interesting, well-written, and well-presented. I really like the study but I am just concerned about the validity of the statistics.

---

## Round 0.4 · accepted · Accept

Thank you for considering the last set of suggestions for your paper and for your response. I am happy to now accept your paper for publication in PeerJ.

When preparing the final version I suggest making the following small edits:

Line 31/32: “for the transfer of DNA”

Line 362/363: “was regarded as statistically significant”